# Optimisation of an nIR-Emitting Benzoporphyrin Pressure-Sensitive Paint Formulation

**DOI:** 10.3390/s25154560

**Published:** 2025-07-23

**Authors:** Elliott J. Nunn, Louise S. Natrajan, Mark K. Quinn

**Affiliations:** 1Department of Chemistry, School of Natural Sciences, University of Manchester, Oxford Road, Manchester M13 9PL, UK; louise.natrajan@manchester.ac.uk; 2Department of Mechanical and Aerospace Engineering, School of Natural Sciences, University of Manchester, Oxford Road, Manchester M13 9PL, UK; mark.quinn@manchester.ac.uk

**Keywords:** pressure-sensitive paints, optical sensors for wind tunnel testing, benzoporphyrins, luminescence, formulation optimization

## Abstract

The use of pressure-sensitive paints (PSPs), an optical oxygen sensing technique, to visualise and measure the surface pressure on vehicle models in wind tunnel testing is becoming increasingly prevalent. Porphyrins have long been the standard luminophore for PSP formulations, with the majority employing the red-emitting platinum(II)-5,10,15,20-tetrakis-(2,3,4,5,6-pentafluorphenyl)-porphyrin. nIR-emitting luminophores, such as Pt(II) tetraphenyl tetrabenzoporphyrins, possess distinct advantages over visible emitting luminophores. In particular, they have wider spectrally useful ‘windows’, facilitating the insertion of a secondary visible emitting temperature-sensitive luminophore to be used for internal calibration without spectral crosstalk that detrimentally impacts PSP performance. In this work, we explore the effect of changing the loading quantity of an nIR-emitting *para*-CF_3_ Pt(II) benzoporphyrin luminophore on the performance of PSP formulations. An optimal luminophore loading of 1.28% wt/wt benzoporphyrin luminophore to polystyrene binder was identified, resulting in a low temperature sensitivity at 100 kPa of 0.61%/K and a large pressure sensitivity at 293 K of 0.740%/kPa. These strong performance metrics, for a polystyrene-based PSP, demonstrate the efficacy of benzoporphyrin luminophores as an attractive luminophore option for the development of a new generation of high-performance PSP formulations that outperform current commercially available ones.

## 1. Introduction

Pressure-sensitive paints (PSPs) are optical oxygen sensors that can facilitate the visualisation of aerodynamic phenomena or measure the full-field surface pressure on vehicle models in wind tunnel testing [1]. A PSP formulation generally consists of a photoactive molecule, known as a luminophore, immobilised in an oxygen permeable binder matrix such as a polymer. The components are dissolved in a solvent and then applied to the surface of a model either by dipping or spraying. During wind tunnel testing, the model is illuminated by a light source, with a wavelength matching the absorption spectrum of the luminophore, which causes it to be excited to the triplet excited state (*T*_1_). From the *T*_1_ excited state, the luminophore can relax in energy back to the ground state through the emission of a photon via phosphorescence. The luminophore *T*_1_ excited state is long enough lived (typically μs) that it can be quenched through collisional quenching by dissolved O_2_ in the permeable binder of the PSP coating, reducing phosphorescence intensity. Due to the aerodynamics of the model being studied, it will experience varying regions of pressure and shear force and thus absorb O_2_ in the PSP binder, resulting in differing levels of phosphorescence quenching across the model. Imaging the varying luminescence intensity of the PSP-coated model during wind tunnel operation can thus provide information on the pressure and aerodynamic phenomena present on the model surface whilst under air flow. PSPs are easy to apply and are non-intrusive to the test model, as opposed to the traditionally used pressure taps, and they have been utilised in a wide variety of measurements, including steady and unsteady flows [2,3,4,5,6,7], rotating flows [8,9,10], cryogenic flows [11,12,13], blast waves [14,15,16], and aeroacoustics [17,18,19].

A key disadvantage of PSP technology currently holding it back from widespread use is a significant temperature sensitivity, in conjunction with the pressure sensitivity, which can severely impact pressure accuracy and thus needs to be corrected for. This temperature sensitivity arises from the non-radiative deactivation of the luminophore, as well as temperature-dependent oxygen diffusion through the permeable binder matrix [1]. A common solution to the temperature sensitivity problem is the use of binary PSP formulations, containing a secondary temperature-sensitive but pressure-insensitive luminophore, which can be used to correct for temperature-induced errors [20,21,22]. However, the two luminophores often experience spectral overlap, reducing overall PSP performance and rendering the two signals difficult to separate [22,23]. Typically, polymer-based PSPs employ the luminophore platinum(II)-5,10,15,20-tetrakis-(2,3,4,5,6-pentafluorphenyl)-porphyrin (PtTFPP) due to its reasonable quantum yield of emission and high photostability [24,25]. However, PtTFPP emits in the visible range (~650 nm), which overlaps with the emission spectra of many desirable secondary temperature-sensitive luminophores.

The use of nIR-emitting luminophores can reduce spectral overlap with secondary luminophores by shifting the pressure-sensitive luminophore emission signal to >700 nm, creating a larger ‘spectral window’ for a secondary luminophore emission signal. Khalil et al. developed the first nIR-emitting PSPs, utilising platinum(II)-meso-tetra(pentafluouorophenyl)porpholactone (PtTFPL), which emits at ~730 nm [21,26]. Using PtTFPL as the pressure-sensitive luminophore, and MgTFPP, which emits at 650 nm, as the temperature-sensitive luminophore, they developed a binary PSP with a very low corrected temperature sensitivity of 0.1%/K. Inspired by their work, we recently developed novel nIR-emitting PSPs using Pt(II) and Pd(II) tetraphenyl tetrabenzoporphyrins, which emit at ~780 nm [27]. These benzoporphyrin PSPs (BP-PSPs) possessed significantly reduced temperature sensitivities (~50%) and much higher brightnesses (~5× higher) compared to traditional PtTFPP PSPs. However, we observed a large decrease in BP-PSP performance with benzoporphyrin luminophore loadings of 3.2% wt/wt benzoporphyrin compared to 0.64% wt/wt in a fluoro/iso/butyl (FIB) polymer-based formulation. In 2005, Grenoble et al. demonstrated that different porphyrin luminophores can possess vastly different concentration-dependent effects on the performance of the resulting PSP formulation [28]. Consequently, this work examines the effect of varying the benzoporphyrin loading on the performance of BP-PSP formulations, in an effort to optimise their performance (Figure 1).

## 2. Materials and Methods

### 2.1. Synthesis and Characterisation

The synthesis details and characterisation data for the benzoporphyrin luminophore, **Pt-*p*CF_3_-BP**, were reported in our previous study [27].

### 2.2. PSP Formulations

The PSP formulations consisted of 4% wt/vol polystyrene to chloroform with varying luminophore loadings of **0.16% wt/wt**, **0.32% wt/wt**, **0.64% wt/wt**, **1.28% wt/wt,** and **2.56% wt/wt** luminophore to polystyrene. The polystyrene was purchased from Sigma Aldrich, Dorset, UK (Mw ~380,000 kDa).

### 2.3. Luminescence Spectra and Lifetimes

Luminescence intensities and lifetime data were recorded on an Edinburgh Instruments (Livingston, UK) FLS-1000 Phosphorescence Lifetime Spectrometer equipped with a 450 W steady-state xenon lamp, a 60 W microsecond pulsed xenon flash lamp (with double 320 mm focal length excitation and emission monochromators in a Czerny–Turner configuration), interchangeable EPL-pulsed diode lasers, and a red-sensitive photomultiplier in Peltier (air-cooled) 53 housing (Hamamatsu Photonics, Hamamatsu City, Japan, R928P). The reported luminescence intensities are the total area of the integrated emission spectrum ranged from 700 to 950 nm. Plotting, fitting, and analysis of data were carried out using Origin 2022b. All data were fitted with exponential decay models and the goodness of fit evaluated by the residual, χ^2^, and R^2^ analysis. Samples were prepared by drop casting approximately 100 μL of the PSP solutions onto a glass microscope slide and allowing them to dry in air. These samples were subsequently taken into an argon-filled glove box to remove all residual volatiles and sealed with another glass slide using vacuum grease around the edges to prevent the diffusion of oxygen into the sample. Lifetimes were also recorded in air-saturated conditions. The reported data are an average of three independent measurement

### 2.4. PSP Performance Studies (A Priori Calibrations)

The BP-PSP formulations were sprayed onto ambersil matt white RAL 9010 base-coated aluminium coupons, using a spray gun, in ten light coats. Freshly sprayed samples were left to air dry for 30 min. The average thickness of the PSP samples was 18 μm, which was determined using an ATP (Ashby-de-la-Zouch, UK) ADT-156 data logging coating thickness metre.

The performance of the PSP formulations was investigated in the standard *a priori* calibration approach. Detailed procedures for the a priori calibration were previously published [29]. Illumination was provided by an air-cooled Lumixtar 30 W 430 nm high-power LED, which was allowed to reach thermal equilibrium for five minutes. Images were captured using a Teledyne Dalsa (Waterloo, ON, Canada) Genie Nano M1920 camera with a 12 mm F1.2 lens fitted with a 610 nm long pass filter.

The luminescence intensity was recorded every 10 kPa from 10 to 150 kPa at 273, 293, and 313 K. The reference luminescence intensity was divided by the luminescence intensity at each pressure and temperature (I_ref_/I) and plotted against the pressure divided by the reference pressure (P/P_ref_) using the modified Stern–Volmer in Equation (1) [1].(1)$$\frac{I_{ref}}{I}=B(T)\frac{P}{P_{ref}}+C(T)$$

The reference luminescence intensity (I_ref_) is the luminescence intensity at 100 kPa at a given temperature, and the reference pressure (P_ref_) is 100 kPa. B(T) and C(T) are calibration constants at a given temperature. The pressure sensitivity at a given temperature, S_p_(T), is calculated using Equation (2) [30].(2)$$S_{p}\left(T\right)={\frac{d(\frac{I_{ref}}{I})}{d(\frac{P}{P_{ref}})}}_{P=P_{ref}}=B\left(T\right)$$

To determine the temperature response of the BP-PSPs, the luminescence intensity was divided by a reference luminescence intensity (I/I_ref_) and plotted against temperature. I_ref_ was the luminescence intensity at 293 K for a given pressure. The response was best fitted with a 2nd order polynomial (Equation (3)):(3)$$\frac{I}{I_{ref}}=C(P)+B_{1}\left(P\right)T+B_{2}\left(P\right)T^{2}$$

C(P), B_1_(P), and B_2_(P) are calibration constants for a given pressure.

The temperature sensitivity at given pressure S_T_(P) for each BP-PSP is calculated as the slope of the plot, from Equation (3), at T = 293 K using Equation (4).(4)$$S_{T}\left(P\right)={\frac{d(\frac{I}{I_{ref}})}{dT}|}_{\;T=T_{293K}\;}={|B}_{1}\left(P\right)+{2B}_{2}\left(P\right)T|$$

The photodegradation (PD) of the PSPs was calculated as the percentage intensity of the original luminescence intensity of the PSP after 45 min of constant illumination at room temperature and pressure (Equation (5)). The light source was positioned 30 cm away from the samples.(5)$$PD=\frac{I_{t=45mins}}{I_{t=0mins}}*100$$

## 3. Results

### 3.1. Luminescene Intensity

The average integrated normalised emission intensity and luminescence spectra for each benzoporphyrin loading BP-PSP across three repeat samples are presented in Figure 2a,b. The integrated emission intensity was calculated by integrating each emission spectrum from 700 to 950 nm. The individual spectra are normalised to the peak height of the **0.64% wt/wt** benzoporphyrin loading BP-PSP sample. The emission intensity increases with increasing benzoporphyrin loading up to **0.64% wt/wt** and then decreases when further increasing the benzoporphyrin loading to **1.28% wt/wt** and **2.56% wt/wt**.

### 3.2. Luminescene Lifetime, τ

The luminescence lifetimes for the thin-film benzoporphyrin loading BP-PSPs in the absence of oxygen, τ_0_, and in air, τ_air,_ are presented in Table 1. τ_0_ remains consistent, at ~56 μs, from a **0.16% wt/wt** to **1.28% wt/wt** benzoporphyrin loading. Upon increasing the benzoporphyrin loading to **2.56% wt/wt**, τ_0_ is found to slightly shorten to 52.9 μs. τ_air_ follows a similar trend to τ_0_ but was best fitted with a bi-exponential decay compared to a mono-exponential decay for τ_0_.

### 3.3. Pressure Sensitivity, S_p_(T)

The pressure sensitivity at a given temperature, S_p_(T), is defined as the change in I_ref_/I against P/P_ref_ at a specific temperature (Equation (2)). I_ref_ and P_ref_ are the luminescence intensity and pressure at 100 kPa for a given temperature. The average pressure responses for each benzoporphyrin loading BP-PSP across three samples were measured at 273, 293, and 313 K. S_p_ (273 K), S_p_ (293 K), and S_p_ (313 K) can be found in Table 2. The modified Stern–Volmer plots for each benzoporphyrin loading can be found in the accompanying Appendix A. The pressure sensitivity increases with increasing temperature due to the temperature dependency of oxygen diffusion through the polymer binder matrix. Therefore, at higher temperatures, there are more O_2_ quencher molecules present in the polymer matrix.

### 3.4. Temperature Sensitivity, S_T_

The luminescence response to temperature for the BP-PSPs was best fitted using a 2nd order polynomial (Equation (3)). The temperature sensitivity at a given pressure S_T_(P) is defined as the change in I/I_ref_ against temperature at 293 K (Equation (4)). The average temperature sensitivity for each benzoporphyrin loading BP-PSP, across three samples, at 100 kPa, S_T_(100 kPa), can be found in Table 3 and is a standard metric for performance comparison. Example plots of the response of I/I_ref_ to increasing temperature at 100 kPa for each benzoporphyrin loading BP-PSP can be found in the accompanying Appendix A.

### 3.5. Photodegradation, PD

The amount of photodegradation, PD, is reported as the percentage of the original luminescence intensity after 45 min of constant illumination at room temperature and pressure. The amount of PD for each benzoporphyrin loading BP-PSP can be found in Table 4.

## 4. Discussion

### 4.1. The Effect of Benzoporphyrin Loading on Luminescence Intensity

The luminescence intensity of a PSP formulation is an important performance metric, and the amount of luminophore loading has previously been demonstrated to have a large impact on the luminescence intensity [28,30]. A brighter PSP emission results in a larger signal-to-noise ratio, rendering data acquisition easier. The luminescence intensity of a PSP can be defined as the product of the molar absorptivity (ε) and quantum yield of emission (Φ) of the employed luminophore species [32]. Ideally, a PSP formulation would use both a highly absorbing and brightly emitting luminophore, reducing the overall amount of luminophore required for optimal sensing. Our chosen luminophore, **Pt-*p*CF_3_-BP,** has a large oxygen-free Φ of 0.63 and a ε of 271,400 M^−1^ cm^−1^ for the Soret band peak at 427 nm in chloroform, which aligns with the wavelength of our excitation source [27]. The shape and position of the emission spectrum of the BP-PSPs was found to remain roughly constant with increasing benzoporphyrin loading. The luminescence intensity of the BP-PSPs initially increases with increasing benzoporphyrin loading. For example, the luminescence intensity increased by a factor of three from a **0.16% wt/wt** to **0.64% wt/wt** benzoporphyrin loading (Figure 2). Increasing the benzoporphyrin loading further to **1.28% wt/wt** results in a 20% decrease in the luminescence intensity from that of the **0.64% wt/wt** loading. When the benzoporphyrin loading is further increased to **2.56% wt/wt,** an even larger 40% reduction in luminescence intensity is observed from that of the **0.64% wt/wt** benzoporphyrin loading. The initial increase in luminescent intensity derives from the increasing concentration of luminescent species, resulting in more luminescence. The reduction in luminescent intensity after the **0.64% wt/wt** benzoporphyrin loading implies that triplet–triplet self-quenching occurs at these higher dye loadings, reducing the overall luminescent intensity.

Considering these results, to optimise the luminescence intensity and therefore the signal of polystyrene-based BP-PSP formulations, we would advise a benzoporphyrin loading of **0.64% wt/wt**. However, if large volumes of BP-PSP are required, for example, to spray a large model, a **0.32% wt/wt** loading could potentially be formulated to use less luminophore and reduce costs at relatively minimal detriment to the resulting luminescence intensity.

### 4.2. The Effect of Benzoporphyrin Loading on the Luminescence Lifetime, τ_0_

The luminescence lifetime in the absence of a quencher species (e.g., O_2_), τ_0_, is another important parameter for PSP formulations. The τ_0_ of the luminophore emission is related to the lifetime of the luminophore *T*_1_ excited state. Using the Stern–Volmer constant (K_SV_ = kqτ_0_), it can be inferred that a longer-lived luminophore *T*_1_ excited state has more probability of undergoing collisional quenching with dissolved O_2_ in the binder matrix, increasing the amount of quenching. Increasing the benzoporphyrin loading has little effect on the τ_0_ of the PSP, which remains constant at ~56.0 μs, up to a benzoporphyrin loading of **1.28% wt/wt** (Table 1, Figure 3).

Upon increasing the benzoporphyrin loading to **2.56% wt/wt**, τ_0_ shortens to 52.9 μs, which has been seen in other benzoporphyrin-based sensor platforms [33]. This shortening of τ_0_ for the **2.56% wt/wt** benzoporphyrin loading aligns well with the observed trends in the luminescence intensity, which is reduced at this higher benzoporphyrin loading. τ_0_ begins to shorten at these higher benzoporphyrin loadings due to the same reason the luminescence intensity decreases—increased triplet–triplet self-quenching. τ_air_ of the different loading BP-PSP formulations follows a similar trend to that of τ_0;_ however, it is much shorter due to the presence of luminescence-quenching O_2_ molecules. Additionally, τ_air_ is best fit using a bi-exponential decay, due to the microheterogeneity of the polymer matrix creating different local microenvironments, which are more and less accessible by the quencher O_2_ molecules [34]. This microheterogeneity leads to a long- and short-lived component, both contributing to τ_air_.

### 4.3. The Effect of Benzoporphyrin Loading on the Pressure Sensitivity, S_p_(T)

The S_p_ at a given temperature, S_p_(T), is another important performance metric for PSP formulations. A larger S_P_ allows for smaller changes in pressure to be resolved. There are two primary factors that affect the S_p_ of a given PSP. As mentioned previously, the τ_0_ of the luminophore species can determine the S_p_ of a PSP formulation through the Stern–Volmer relation. Additionally, the permeability of the binder matrix determines the concentration of dissolved O_2_ quencher molecules at a given pressure [1]. Therefore, a more air permeable binder matrix generally facilitates more luminescence quenching and increases the S_p_(T) of a PSP formulation. In addition to these two factors, the luminophore-binder compatibility can significantly affect the S_p_(T) of a given PSP. We recently demonstrated that -CF_3_-bearing porphyrins and benzoporphyrins significantly increase the S_p_(T) of polymer-based PSPs, compared to the standard -F-bearing PtTFPP luminophore [27,29]. The S_P_(273 K) of the different benzoporphyrin loading BP-PSP formulations (Table 2 and Figure 4) steadily increases with increasing benzoporphyrin loading from 0.635%/kPa at **0.16% wt/wt** to 0.717%/kPa at **1.28% wt/wt**.

Increasing the benzoporphyrin loading to **2.56% wt/wt** from **1.28% wt/wt**, akin to the luminescence intensity and τ_0_, has a negative effect, reducing the Sp(273 K) to 0.694%/kPa, which likely derives from the slightly reduced τ_0_ of this formulation. Changing the benzoporphyrin loading has no effect on the linearity of the highly linear modified Stern–Volmer responses, with an R^2^ = 0.999 across the different BP-PSP formulations.

At higher temperatures, S_p_(T) increases due to the temperature dependency associated with O_2_ diffusion through the permeable binder matrix (Table 2). An ideal PSP formulation would possess a high S_p_(T) that is constant across a wide temperature range. S_p_(T) is found to be affected the least by temperature for benzoporphyrin loadings of **1.28% wt/wt**; for example, increasing the temperature from 273 K to 313 K increases S_p_(T) by 4%. S_p_ is found to be affected the most by temperature for benzoporphyrin loadings of **0.16% wt/wt**; for example, increasing the temperature from 273 K to 313 K increases S_p_(T) by 13%.

In light of these results, using a benzoporphyrin loading of **1.28% wt/wt** would be advisable to achieve an optimal S_p_(T) for BP-PSP formulations that is the least sensitive to temperature.

It is useful to compare the optimised BP-PSP formulations to other widely used formulations to demonstrate their efficacy as new PSPs. For example, the S_p_(293 K) for the **1.28% wt/wt** benzoporphyrin loading BP-PSP (0.74%/kPa) is much greater than the commercially available polymer-based PSP UniCoat PSP (0.5%/kPa) and slightly higher than the high-performance UniFIB formulation (0.7%/kPa) [35]. This exceptional S_p_(293 K) for a polystyrene-based PSP makes this newly optimised BP-PSP an attractive option for a new generation of high-performance optical pressure sensors.

### 4.4. The Effect of Benzoporphyrin Loading on the Temperature Sensitivity, S_T_(P)

A high S_T_ can severely impact the pressure accuracy of a PSP formulation, and therefore, a high-performance PSP would ideally have zero temperature dependence. We recently demonstrated that BP-PSP formulations exhibit a significantly reduced S_T_ (~50%) compared to traditional PtTFPP-based PSP formulations, rendering them excellent candidates for a new generation of high-performance PSPs [27]. We theorised that this reduction in S_T_ when using benzoporphyrin luminophores compared to PtTFPP is due to an increased luminophore-binder compatibility from the -CF_3_ substituents. Changing the benzoporphyrin loading has a large effect on the S_T_ of the resulting BP-PSP formulation (Table 3). Initially, the S_T_(100 kPa) is found to steadily decrease with increasing benzoporphyrin loading; for example, increasing the benzoporphyrin loading from **0.16% wt/wt** to **1.28% wt/wt** decreases the overall S_T_(100 kPa) by 40% from 1.03%/K to 0.61%/K. Further increasing the benzoporphyrin loading to **2.56% wt/wt** is found to increase the S_T_(100 kPa) to 0.73%/K.

Considering these results, we advise the use of a **1.28% wt/wt** benzoporphyrin loading in future BP-PSP formulations to achieve an optimally low S_T_(100 kPa) of 0.61%/K. If we compare this optimised BP-PSP formulation to ISSI UniCoat, with a S_T_(100 kPa) of 1.3%/K, a 56% reduction in S_T_(100 kPa) can be achieved using BP-PSP over traditional PtTFPP-based PSPs [35]. Indeed, the low S_T_(100 kPa) of the **1.28% wt/wt** loading BP-PSP is comparable to that of the high-performance ISSI UniFIB PSP, which has an S_T_(100 kPa) of 0.4%/K [35]. However, the ISSI UniFIB formulation utilises the low S_T_ binder fluoro/iso/butyl (FIB) polymer, which has previously been demonstrated to greatly reduce S_T_(100 kPa) [29,36]. In contrast, our optimised BP-PSP uses a traditionally high S_T_ binder, polystyrene, and thus, it is remarkable that an S_T_(100 kPa) comparable to that of the ISSI UniFIB can be achieved using this newly optimised BP-PSP formulation.

It is also important to consider the sensitivity of S_T_(P) to pressure (Figure 5) for a given PSP formulation. An ideal PSP formulation would possess a minimal S_T_ that is insensitive to pressure. We demonstrated that BP-PSPs exhibit a greatly reduced sensitivity of S_T_(P) to pressure compared to traditional PtTFPP-based PSP formulations. Like the trend in S_T_(100 kPa), the sensitivity of S_T_(P) to pressure initially decreases (becomes shallower on the plots in Figure 5), going from a benzoporphyrin loading of **0.16% wt/wt** up to **1.28% wt/wt**. Further increasing the benzoporphyrin loading to **2.56% wt/wt** increases the sensitivity of S_T_(P) to pressure, thus reducing the overall sensor performance. The S_T_ at 10 kPa, S_T_(10 kPa), for all the benzoporphyrin loadings is small and relatively constant, which is expected because at low pressures, the non-radiative decay of the luminophore excited state is the dominant contributor to the S_T_ of a PSP formulation [1]. However, at higher pressures, where the temperature dependency of oxygen diffusion through the binder becomes the dominant contributor to the S_T_ of a given PSP, S_T_ deviates greatly with increasing benzoporphyrin loading. It is currently unclear why the sensitivity of S_T_(P) to pressure is so heavily affected by the amount of benzoporphyrin loading for BP-PSP formulations.

### 4.5. The Effect of Benzoporphyrin Loading on PD

The PD of PSP formulations is an unavoidable consequence of the photoexcitation of the luminophore molecules into the *T*_1_ excited state. Over long periods of exposure time, PD results in a deviation of the measured intensity ratios from the calibrated intensity ratios, leading to inaccurate pressure measurements. PD occurs via two mechanisms [37]. Firstly, quenching of the luminophore *T*_1_ excited state by ground state ^3^O_2_ results in the formation of the highly reactive and thus highly damaging ^1^O_2_, singlet oxygen radical species. The generated ^1^O_2_ can then react with the luminophore molecule directly, damaging it through photooxidation, or with the binder matrix, generating more radical species that propagate and damage the luminophore molecule. In addition to photooxidation, the absorption of a photon by a luminophore molecule can result in bond rupture, which degrades the luminophore. Both processes result in an overall decrease in the luminescence intensity as luminophore molecules degrade. The amount of PD is unaffected by the benzoporphyrin loading (Table 4 and Figure 6), with a consistent 1% reduction in the original luminescence intensity after 45 min of constant illumination at room temperature and pressure across the different benzoporphyrin loading BP-PSPs.

## 5. Conclusions

In conclusion, we have altered the benzoporphyrin luminophore loading from **0.16% wt/wt** to **2.56% wt/wt**, for a benzoporphyrin pressure-sensitive paint (BP-PSP) formulation, to identify the benzoporphyrin loading that affords optimal PSP performance. The effect of changing the benzoporphyrin loading on the luminescence intensity, emission lifetime (τ), pressure sensitivity at a given temperature (S_p_(T)), temperature sensitivity at a given pressure (S_T_(P)), and photodegradation (PD) was examined to evaluate the overall BP-PSP performance. It was found that increasing the benzoporphyrin loading initially increased the luminescence intensity, up to an optimal loading of **0.64% wt/wt**, after which the luminescence intensity decreased due to the occurrence of triplet–triplet self-quenching. However, the reduction in luminescence intensity when increasing the benzoporphyrin loading from **0.64% wt/wt** to **1.28% wt/wt** was relatively small. Increasing the benzoporphyrin loading afforded a constant τ_0_ of ~56 μs up to a **1.28% wt/wt** loading. When the benzoporphyrin loading was further increased to **2.56% wt/wt,** a shortening of τ_0_ to 52.9 μs was observed. The S_p_(273 K) of the BP-PSPs steadily increased with increasing benzoporphyrin loading, up to an optimal loading of **1.28% wt/wt**, affording a relatively high S_p_(273 K) of 0.717%/kPa. Further increasing the benzoporphyrin loading to **2.56% wt/wt** reduced S_p_(273 K) to 0.694%/kPa, which is likely due to the reduction in lifetime because of the increased triplet–triplet self-quenching at these higher loadings. S_T_(100 kPa) was found to decrease with increasing benzoporphyrin loading, up to an optimal loading of **1.28% wt/wt**, which afforded a very low S_T_(100 kPa) of 0.61%/K for a polystyrene-based PSP. Further increasing the benzoporphyrin loading to **2.56% wt/wt** increased the S_T_(100 kPa), reducing sensor performance. Additionally, like the trends in S_T_(100 kPa), increasing the benzoporphyrin loading decreased the sensitivity of S_T_(P) to pressure, with the **1.28% wt/wt** benzoporphyrin loading affording a BP-PSP with a S_T_(P) that was least sensitive to pressure. This large decrease in the sensitivity of S_T_(P) to pressure is an important step towards higher performing PSP formulations which can maintain high performance over a large temperature range. The PD was found to be constant with increasing benzoporphyrin loading, with an observed 1% reduction in luminescence intensity after 45 min of constant illumination at room temperature and pressure for all the BP-PSPs.

Overall, the loading of the benzoporphyrin luminophore in BP-PSP formulations was found to have a large effect on the luminescence intensity, S_p_(T), and S_T_(P). An optimal benzoporphyrin loading of **1.28% wt/wt** affords a high S_p_ and low S_T_ whilst maintaining a relatively high luminescence signal. These results demonstrate the efficacy of BP-PSPs as a new generation of high-performance PSPs, and we hope they will assist in the development of increasingly higher performing formulations.

## Figures and Tables

**Figure 1 sensors-25-04560-f001:**
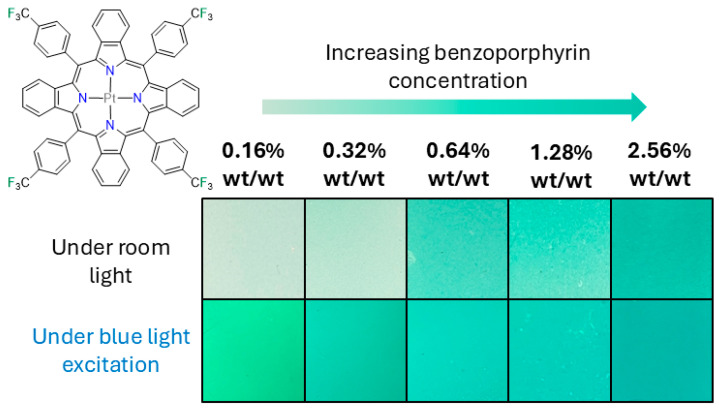
The chemical structure of the benzoporphyrin luminophore, **Pt-*p*CF_3_-BP**, employed in this study, and photographs of the different luminophore loading BP-PSPs under room light and under blue excitation light (430 nm) whilst using a 490 nm long pass filter attached to a smartphone.

**Figure 2 sensors-25-04560-f002:**
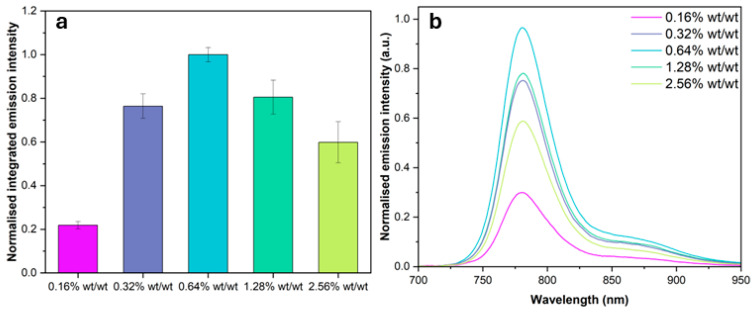
(**a**) The average luminescence intensity of the different benzoporphyrin loading BP-PSP formulations in the absence of oxygen with associated % errors across three repeat samples for each loading. (**b**) The average emission spectrum of the different benzoporphyrin loading BP-PSP formulations in the absence of oxygen.

**Figure 3 sensors-25-04560-f003:**
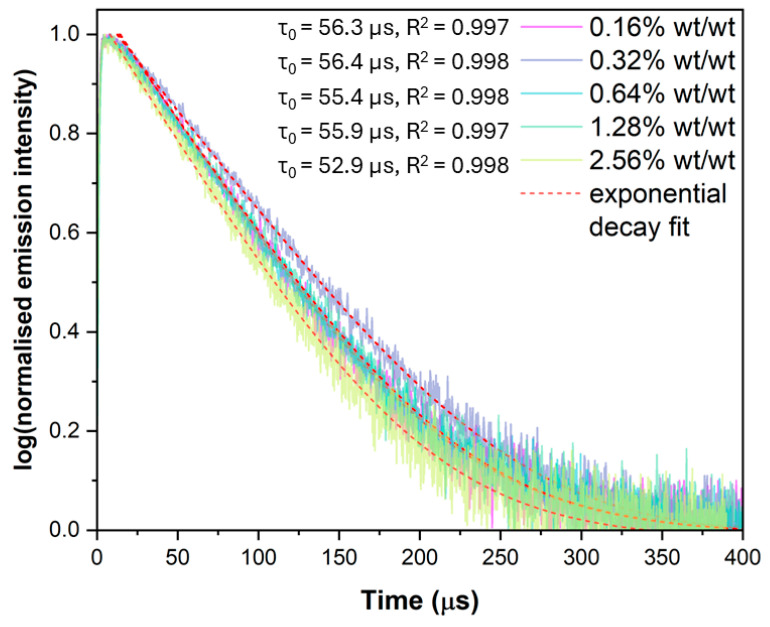
Lifetimes of emission, in the absence of oxygen, for the different benzoporphyrin loading BP-PSP formulations, with the associated mono-exponential decay fits.

**Figure 4 sensors-25-04560-f004:**
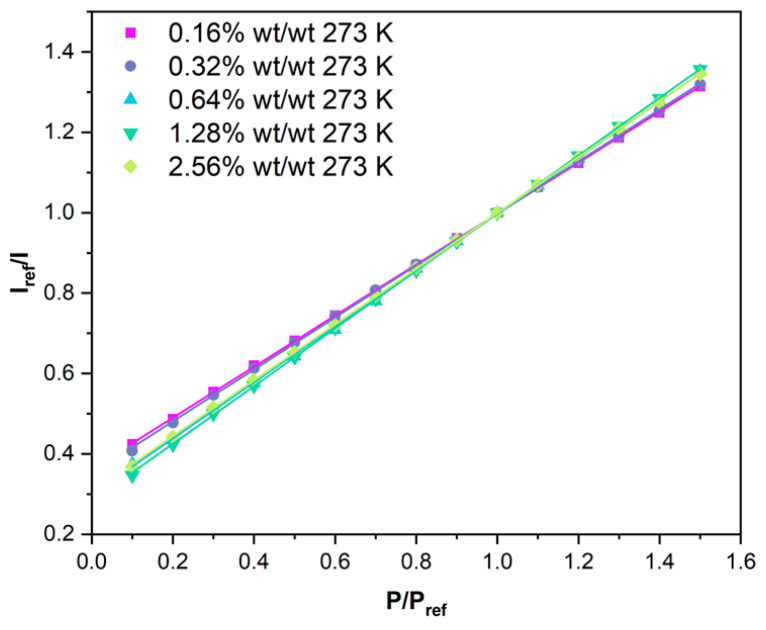
The modified Stern–Volmer (Equation (2)) luminescence responses to pressure for the different benzoporphyrin loading BP-PSP formulations at 273 K, with their associated linear fits. I_ref_ is the luminescence intensity at 100 kPa and 273 K, and P_ref_ is 100 kPa for each benzoporphyrin loading. The data for the **0.64% wt/wt** benzoporphyrin loading were reported previously in a different format [27].

**Figure 5 sensors-25-04560-f005:**
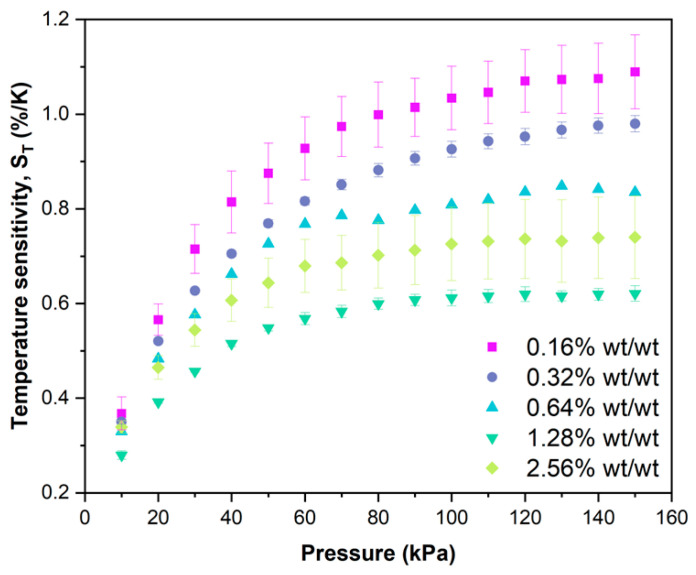
The change in S_T_ with increasing pressure for the different benzoporphyrin loading BP-PSP formulations. The data for the **0.64% wt/wt** benzoporphyrin loading were reported previously in a different format [27].

**Figure 6 sensors-25-04560-f006:**
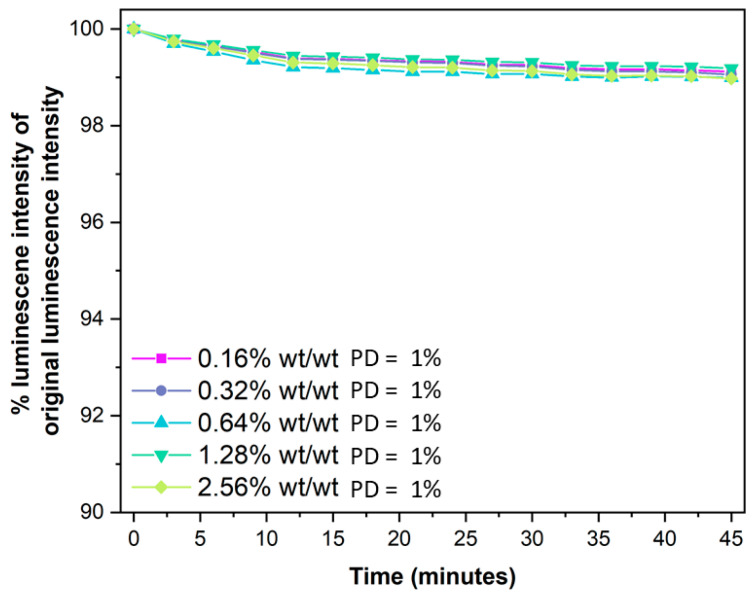
The % luminescence intensity of the original luminescence intensity, at time = 0 min, every 3 min for 45 min. The overall photodegradation (PD) is calculated using Equation (5).

**Table 1 sensors-25-04560-t001:** The luminescence lifetime in the absence of oxygen, τ_0_, and in air, τ_air_, for the thin-film BP-PSP samples.

Benzoporphyrin Loading	τ_0_ (μs)	τ_air_ * (μs)
**0.16% wt/wt**	56.3	14.9
**0.32% wt/wt**	56.4	14.9
**0.64% wt/wt**	55.4	14.5
**1.28% wt/wt**	55.9	14.1
**2.56% wt/wt**	52.9	13.0

* τ_air_ was best fitted with a bi-exponential decay equation and so is reported as the intensity average lifetime, which is calculated as τ_air_ = (B_1_τ_1_)^2^ + (B_2_τ_2_)^2^/(B_1_τ_1_) + (B_2_τ_2_) [31], where B_x_ and τ_x_ are the initial intensity and lifetime of component x.

**Table 2 sensors-25-04560-t002:** The pressure sensitivity at a given temperature, S_p_(T), for the different BP-PSP formulations at 273 K, 293 K, and 313 K, with associated standard errors.

Benzoporphyrin Loading	S_p_(T) (%/kPa)		
	T = 273 K	T = 293 K	T = 313 K
**0.16% wt/wt**	0.635 ± 0.006	0.685 ± 0.006	0.716 ± 0.003
**0.32% wt/wt**	0.647 ± 0.003	0.693 ± 0.004	0.724 ± 0.001
**0.64% wt/wt ^a^**	0.699	0.737	0.751
**1.28% wt/wt**	0.717 ± 0.001	0.740 ± 0.004	0.748 ± 0.002
**2.56% wt/wt**	0.694 ± 0.018	0.713 ± 0.007	0.743 ± 0.006

^a^ data previously published in a different format [27].

**Table 3 sensors-25-04560-t003:** The average S_T_(100 kPa) for each benzoporphyrin loading BP-PSP, with associated standard errors across three repeated samples for each loading.

Benzoporphyrin Loading	Temperature Sensitivity at 100 kPa S_T_(100 kPa) (%/K)
**0.16% wt/wt**	1.03 ± 0.07
**0.32% wt/wt**	0.93 ± 0.02
**0.64% wt/wt** ^a^	0.81
**1.28% wt/wt**	0.61 ± 0.02
**2.56% wt/wt**	0.73 ± 0.08

^a^ data previously published in a different format [27].

**Table 4 sensors-25-04560-t004:** The amount of PD after 45 min of constant illumination at room temperature and pressure for the different benzoporphyrin loading BP-PSPs.

Benzoporphyrin Loading	Photodegradation PD (%)
**0.16% wt/wt**	1
**0.32% wt/wt**	1
**0.64% wt/wt**	1
**1.28% wt/wt**	1
**2.56% wt/wt**	1

## Data Availability

The Raw data can be made available upon request.

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
