# Peer review of "Optimisation of an nIR-Emitting Benzoporphyrin Pressure-Sensitive Paint Formulation"

_sensors, 2025, doi:10.3390/s25154560_

Round 1
Reviewer 1 Report
Comments and Suggestions for Authors
The comments to the authors are shown in the attached file.

Author Response
Please see the reply to reviewer 1's comments in the attached document

Reviewer 2 Report
Comments and Suggestions for Authors
In this work, the authors explore the effect of changing the loading quantity of a NIR-emitting para-CF3 Pt(II) benzoporphyrin luminophore on the performance of PSP formulations. This work is very interesting and should be accepted. I only have two questions to communicate with the authors.
- When conducting a wind tunnel laboratory, not only will pressure be generated, but also shear force will be generated, which is not mentioned in the article.
- What is the typical temperature range when conducting a wind tunnel laboratory? In addition, the phosphorescence lifetime of this near-infrared dye is shorter. Do phosphorescent materials with longer lifetimes have better performance?
Author Response
Please see reply to reviewer 2 in the attached document

Round 2
Reviewer 1 Report
Comments and Suggestions for Authors
The comments for the Authors are shown in the attached file.

Author Response
Please find our response to the reviewer comments in the attached document.
